# Understanding and Improving Length Generalization in Recurrent Models

**Ricardo Buitrago Ruiz** [1 2]    **Albert Gu** [1 2]

## Abstract

Recently, recurrent models such as state space models and linear attention have become popular due to their linear complexity in the sequence length. Thanks to their recurrent nature, in principle they can process arbitrarily long sequences, but their performance sometimes drops considerably beyond their training context lengths—i.e. they fail to length generalize. In this work, we provide comprehensive empirical and theoretical analysis to support the *unexplored states hypothesis*, which posits that models fail to length generalize when during training they are only exposed to a limited subset of the distribution of all *attainable* states (i.e. states that would be attained if the recurrence was applied to long sequences). Furthermore, we investigate simple training interventions that aim to increase the coverage of the states that the model is trained on, e.g. by initializing the state with Gaussian noise or with the final state of a different input sequence. With only 500 post-training steps ($\sim 0.1\%$ of the pre-training budget), these interventions enable length generalization for sequences that are orders of magnitude longer than the training context (e.g. $2k \longrightarrow 128k$) and show improved performance in long context tasks, thus presenting a simple and efficient way to enable robust length generalization in general recurrent models.

## 1. Introduction

Recurrent architectures like state space models (Gu et al., 2022; Smith et al., 2023; Gu & Dao, 2023; Dao & Gu, 2024) and linear attention variants (Katharopoulos et al., 2020; Peng et al., 2023; Yang et al., 2024b;c) compress the previous context of a sequence into a state, with each output depending on previous tokens only through the state. In addi-

tion to matching the performance of Transformers (Vaswani et al., 2017) across many tasks, the recurrent mechanism brings two benefits: the ability to efficiently process long sequences thanks to its linear complexity, and the capacity to easily process tokens beyond their training context by simply rolling out the state. Nevertheless, in practice these benefits are often unrealized, given that their performance can drop considerably when the sequence length exceeds their training context (Waleffe et al., 2024; Ben-Kish et al., 2024; Ye et al., 2025; Yuan et al., 2024). This naturally leads to two questions: (1) why do these models fail to length generalize? and (2) how can we efficiently enable length generalization across several recurrent models?

Recently, some works have studied the length generalization of Mamba (Dao & Gu, 2024) and have proposed solutions such as forcing the model to forget previous context (Chen et al., 2024b) or skipping tokens in the state update to reduce the effective context of the processed sequence (Ye et al., 2025; Ben-Kish et al., 2024). However, these methods require changing the internal mechanism of Mamba and might not be easily transferable to other architectures. Other works have linked length generalization to state capacity and overfitting (Wang, 2024; Chen et al., 2024b), proposing training on longer sequences and with Truncated Backpropagation Through Time (TBTT) (Williams & Peng, 1990; Sutskever, 2013) as a way to enable length generalization. In this work, we reason about the distribution of states that the model is trained on to introduce a precise hypothesis that explains why recurrent models fail to length generalize. Moreover, we perform comprehensive interventions that elucidate on what distributions recurrent models need to be trained to enable length generalization.

More concretely, we propose the *unexplored states hypothesis*, which suggests that **models fail to length generalize when their recurrence applied to long sequences produces state distributions that have not been explored during training**. We support this hypothesis through several experiments and additionally introduce Effective Remembrance, a novel metric that measures the impact of previous parts of the context in the output of the model. We find out that models that fail to length generalize are disproportionately impacted by the initial tokens of the sequence, suggesting that these models overfit to states produced early in the sequence when they are trained on short contexts with

[1]Carnegie Mellon University [2]Cartesia AI. Correspondence to: Ricardo Buitrago Ruiz <ricardobuitragoruiz@hotmail.com>.

*Proceedings of the $42^{nd}$ International Conference on Machine Learning*, Vancouver, Canada. PMLR 267, 2025. Copyright 2025 by the author(s).

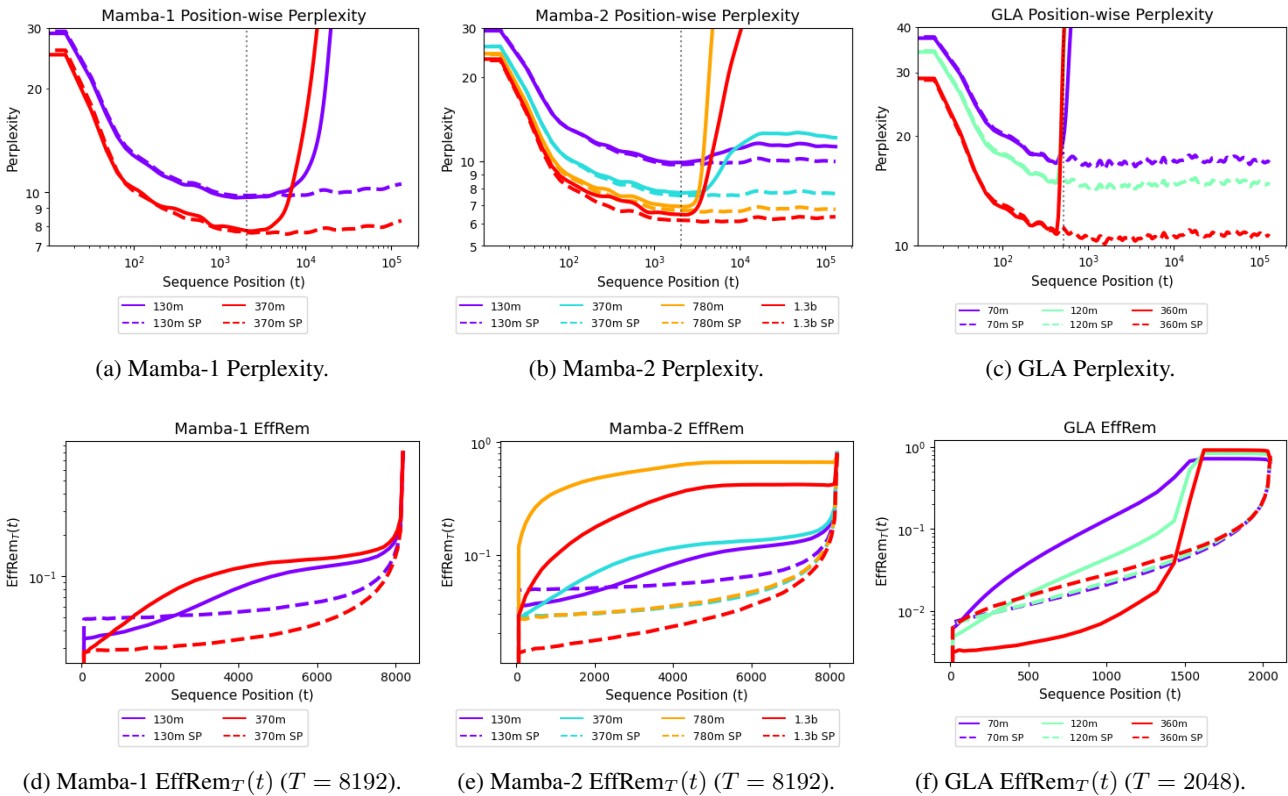

(a) Mamba-1 Perplexity.  (b) Mamba-2 Perplexity.  (c) GLA Perplexity.

(d) Mamba-1 EffRem$_T(t)$ ($T = 8192$).  (e) Mamba-2 EffRem$_T(t)$ ($T = 8192$).  (f) GLA EffRem$_T(t)$ ($T = 2048$).

*Figure 1.* (**Top**) Perplexity as a function of token position on the Pile validation dataset (Gao et al., 2020) for the official Mamba-1 and Mamba-2 checkpoints trained with context $T = 2048$, as well as for Gated Linear Attention (GLA) models trained with context $T = 512$. In dashed lines, we show the same models post-trained with State Passing (SP), which is an intervention that initializes the state with the final state of a different sequence (see Section 4.1). State Passing is a simple technique that enables length generalization across several recurrent architectures. Mamba-2 and GLA are post-trained for 500 steps and Mamba-1 is post-trained for 1000 steps. A similar plot for RWKV-v6 (Peng et al., 2024a) is shown in Figure 10. (**Bottom**) Effective Remembrance for recurrent models and their State Passing post-trained counterparts. Effective Remembrance at time $t$ roughly measures the impact of the tokens at positions $[0, t)$ on the output of the model at a later position $T$, with 0 indicating no impact (no "effective remembrance" of tokens $[0, t)$) and 1 indicating maximal impact (see Section 3.4 for a precise definition). The baseline models are disproportionately affected by tokens that are very far away in the past, indicating that they are not correctly handling the recent context. State Passing fixes this behavior.

a zero-initialized fixed state.

Based on these findings, we explore training interventions that modify the initial state to expose the models to a wider range of state distributions (which is equivalent to starting from some given initial context). We investigate techniques such as TBTT and additionally propose new ones that are directly motivated by our hypothesis, such as initializing the state with some type of Gaussian noise. Through a comprehensive comparison of these interventions, we conclude that the key to length generalization is training on initial states that are similar to the states that the model attains when processing long sequences—in particular, training on long sequences (directly or indirectly through TBTT) is not always necessary. In our experiments, with as little as 500 post-training steps ($\sim 0.1\%$ of the pre-training budget) we enable length generalization from 2k training contexts to sequences of length 128k at validation. Furthermore, the in-

terventions enable length extrapolation in long context tasks such as BABILong (Kuratov et al., 2024), passkey retrieval (Mohtashami & Jaggi, 2023) and synthetic copying (Jelassi et al., 2024); and also change the curves of the Effective Remembrance metric so that the model no longer overfits to the states of early parts of the sequence.

Our study argues that **length generalization in recurrent models is expected to be readily achievable through simple training interventions**. This simplifies the process of comparing new recurrent architectures by allowing researchers to primarily focus on their in-length performance, which we consider particularly significant in light of the recent proliferation of newly proposed recurrent architectures (Peng et al., 2023; Sun et al., 2023; Beck et al., 2024; Katsch, 2023; Ma et al., 2024; Liu et al., 2024; De et al., 2024; Arora et al., 2024; Sun et al., 2024; Yang et al., 2024b;c;a). As a whole, our contributions bring new theoretical insights

on the behavior of stateful models and simple empirical techniques to improve recurrent models at large.

## 2. Preliminaries

### 2.1. Notation

Given a sequence $x = (x_0, x_1, ..., x_T)$ we will use $x_{s:t}$ to refer to the subset of the sequence $(x_s, x_{s+1}, ..., x_t)$. Additionally, for any sequence $x$ we will use $x^\times$ to denote the multiplication of elements of $x$, i.e. $x^\times = \prod_{i=0}^{T} x_i$. For convenience, if $s > t$, we will define $x_{s:t}^\times = 1$.

### 2.2. Recurrent Models

The core of both Mamba and Mamba-2 is a state space model (SSM), which is a transformation of a 1-dimensional sequence $x \in \mathbb{R}^T \to y \in \mathbb{R}^T$ through an implicit latent state $h \in \mathbb{R}^{(T,N)}$. In its general form, it can be written as:

$$h_t = A_t h_{t-1} + B_t x_t \tag{1a}$$
$$y_t = C_t^T h_t \tag{1b}$$

where $A_t \in \mathbb{R}^{(N,N)}$, $B_t \in \mathbb{R}^{(N,1)}$, and $C_t \in \mathbb{R}^{(N,1)}$. The output $y_t$ depends on the past history $x_{0:t}$ only through the state $h_t$. Thus, in autoregressive models $h_t$ maintains a compressed representation of the past $x_{0:t}$ which is useful to predict the next token. Equation 1a is valid for $t \in [0, T]$. For $t = 0$, it is also necessary to define $h_{-1}$, which we denote by *initial state*. In the standard implementations of Mamba, the state $h_{-1}$ is initialized with zeros $h_{-1} = 0$. Note that this is equivalent to starting to predict the sequence $y_{0:T}$ without any previous context.

Throughout this work, we use the term "recurrent models" to refer to architectures that accept a formulation like the one presented in Equation 1, which includes most modern recurrent models like Linear Attention (Katharopoulos et al., 2020), RWKV (Peng et al., 2023), Retnet (Sun et al., 2023) and Gated Linear Attention (Yang et al., 2024b), to name a few. The difference between the architectures mostly resides in the parametrization of $A_t$ $B_t$ and $C_t$; and in how they are obtained from the inputs. We note that some modern recurrent architectures do not accept a formulation like Equation 1 (e.g. xLSTM (Beck et al., 2024)); we hypothesize that they would exhibit a similar behavior but we leave them outside the scope of this work. We refer to Yang et al. (2024c) for an overview of modern recurrent models.

## 3. Analyzing the Length Generalization Failure of Recurrent Models

In this section, we identify training conditions under which models fail to length generalize and explain this behavior through our proposed *unexplored states hypothesis*.

### 3.1. Definition of Length Generalization

First, we provide a concrete definition for position-wise perplexity—i.e, the average perplexity that the model achieves at each position in the sequence.

**Definition 3.1** (Position-wise Perplexity). Let $q(\cdot|c)$ be the next token probabilities of an autoregressive sequential model given a context $c$, and let $\mathcal{D}$ be a distribution over sequences. We define the position-wise perplexity at position $t$ as the average perplexity that the autoregressive model achieves at position $t$:

$$\text{Position-wise Perplexity}(t) = E_{x \sim \mathcal{D}}[\exp(q(x_t|x_{0:t-1})]$$

Position-wise perplexities can easily be computed in a dataset simply by averaging the typical perplexity by sequence position. These values serve to define length generalization:

**Definition 3.2** (Length Generalization). Let $\mathcal{M}$ be an autoregressive sequential model trained with context $T_{\text{train}}$, $[0, T]$ an interval with $T > T_{\text{train}}$, and $\mathcal{D}$ a distribution over sequences. Let $p^\star$ be the minimum position-wise perplexity that $\mathcal{M}$ achieves on $[0, T_{\text{train}}]$, and let $t^\star$ be the position where it is achieved. Then, $\mathcal{M}$ is said to length generalize in $[0, T]$ if the position-wise perplexities for $t \in [t^\star, T_{\text{train}}]$ are smaller or equal than $p^\star$.

### 3.2. Short Training Contexts Impede Length Generalization

In this subsection we show the results for models trained from scratch with different training contexts. In Figure 2 it can be seen that when the training context length is too short, the models' perplexities diverge for positions after the context length. Additionally, the larger the model, the longer the context needed to length generalize. Based on these results, we conclude that *the longer the training context, the better the length generalization*. Details on the training recipe and model configurations are given in Section B.1.

### 3.3. Training for Many Tokens Impedes Length Generalization

In the previous subsection we trained models for several times what Chinchilla scaling laws dictate, which is typical to maximize their performance. Now, we will study whether the failure to length generalize also occurs when training for fewer tokens. Figure 3 shows the position-wise perplexity of a Mamba-2 model at different checkpoints during training, showing that the model needs to be trained for at least 7.5x Chinchilla scaling laws to fail to length generalize. Thus, *extended training hinders length generalization*.

We note that, in addition to short contexts and extended training, other hyperparameters also influence length generalization in our experiments. For example, in Section E we

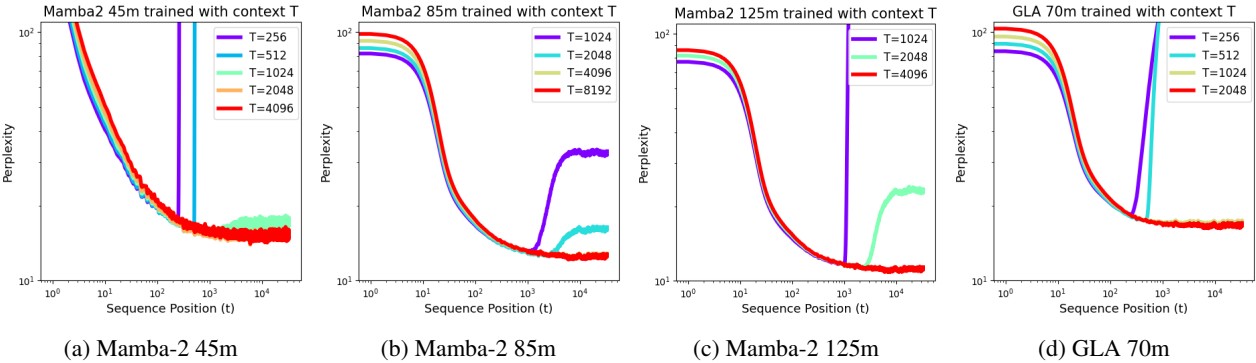

| (a) Mamba-2 45m | (b) Mamba-2 85m | (c) Mamba-2 125m | (d) GLA 70m |

*Figure 2.* Position-wise perplexities for Mamba-2 (Dao & Gu, 2024) and GLA (Yang et al., 2024b) trained from scratch with different context lengths $T$ on the Pile (Gao et al., 2020). The longer the training context, the better the length generalization. The 45m model is trained for 22.5B tokens (25x Chinchilla laws), the 70m and 85m are trained for 34B tokens (20x Chinchilla Laws), and the 125 model is trained for 25B tokens (10x Chinchilla Laws).

observe that having an extended learning rate warm-up period hinders length generalization. This indicates that there is a complex relationship between length generalization and training recipes. In Section 4 we will show that there are inexpensive training interventions that enable length generalization, removing the need to carefully tune the training recipe to achieve generalization.

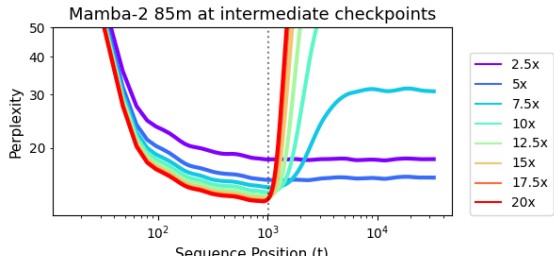

*Figure 3.* Position-wise perplexities of a Mamba-2 85m model trained for different number of tokens with a context of 1024 on the Pile (Gao et al., 2020). 2.5x means that the model is trained for 2.5 times what Chinchilla scaling laws dictate for that model size. Thus, the checkpoints correspond to a range between 4.25B and 34B tokens. In this case, the failure to length generalize occurs after training for more than 7.5x Chinchilla laws.

### 3.4. Effective Remembrance: a Metric to Understand How Sequence Models Process Context

Sequence models would perform reasonably well on long sequences if they only used a sliding context window of length equal to the training context to predict each output, as this would be equivalent to processing sequences of the same length as those seen during training. This is the mechanism of some architectures like Sliding Window Attention (Beltagy et al., 2020). However, recurrent models use the the full context for their predictions (indeed, there is not a

straightforward way to eliminate previous parts of the sequence from the state). Consequently, their failure to length generalize arises because their handling of recent context is compromised by having already processed earlier parts of the sequence.

To analyze this phenomenon further, we propose comparing the outputs of the model when it is given the full context, and the output when it is given only a recent context. Concretely, given a context $x_{0:T}$, an autoregressive model predicts a vector of probabilities for the next token position, $q(\cdot|x_{0:T}) \in \mathbb{R}^{|\mathcal{V}|}$, where $|\mathcal{V}|$ is the vocabulary size. We propose Effective Remembrance as a measure to understand how the output of different partial contexts $x_{t:T}$ differs from the output of the full context.

**Definition 3.3** (Effective Remembrance). Given an autoregressive model which outputs probabilities $q(\cdot|\text{context})$ over a vocabulary $\mathcal{V}$, an input sequence $x_{0:T}$ and a distance between probability distributions $d$, we define the Effective Remembrance for $t \in [0, T]$ as:

$$\text{EffRem}_T(t) = d(q(\cdot|x_{0:T}), q(\cdot|x_{t:T})) \qquad (2)$$

Unless otherwise stated, we will use total variation[1] $d = \text{TV}$ as a distance between distributions:

$$\text{TV}(q(\cdot), q'(\cdot)) = \frac{1}{2} \sum_{x \in \mathcal{V}} |q(x) - q(x')| \qquad (3)$$

Additionally, we will present the values of Effective Remembrance averaged over a dataset. We note that Effective Remembrance is applicable to all autoregressive sequence models, not just recurrent ones.

---

[1]In Section G we perform an ablation on the choice of the distance metric for Effective Remembrance and conclude that the shape of the curve $\text{EffRem}_T(t)$ is very similar when using total variation, the Jensen-Shannon distance or cosine similarity.

Effective Remembrance can be understood as how much the past tokens $x_{0:t-1}$ influence the output at time $T$. If $\text{EffRem}_T(t) = 0$, this means that the predictions using $x_{t:T}$ and using $x_{0:T}$ are the same, meaning that the model does not "effectively remember" any of the past tokens $x_{0:t-1}$. Conversely, if $\text{EffRem}_T(t)$ is high, the model is substantially influenced by the tokens $x_{0:t-1}$, since removing them from the context changes the prediction significantly. Naturally, for distribution such as text we would expect $\text{EffRem}_T(t) \approx 0$ for $t \ll T$ (tokens that are very far away from $T$ barely have an effect on the prediction), and $\text{EffRem}_T(t) \longrightarrow 1$ when $t \longrightarrow T$.[2]

Figure 1 shows the Effective Remembrance of several recurrent models, using $T$ four times larger than the training context (8192 for Mamba and 2048 for GLA). Models that fail to length generalize are substantially affected by far away tokens. Indeed, the 1.3b Mamba-2 model has high values of $\text{EffRem}_T(t)$ for small $t$, indicating that its outputs strongly depend on the initial tokens (in other words, if the initial tokens were not included for the prediction, the output would be very different). Therefore, the failure to length generalize is linked to the model depending too much on the initial tokens and overfitting to the states that arise when processing the early parts of the sequence without previous context (i.e. with a fixed zero-initialized state).

### 3.5. The Distribution of States Changes Over Time

We can gain further insights into why recurrent models fail to length generalize by studying statistics of the distribution of the state *over time*. Concretely, if we have a distribution of sequences of arbitrary length $X$, we can study the distribution of the state at time $t$, $\mathcal{D}_t := h_t(X_{0:t})$, where $h_t(x)$ refers to rolling out the state recurrence (e.g. Equation 1a) on $x$. In Figure 4 we show that the standard deviation of the state of Mamba-2 130m increases over time. In particular, after the training context the model encounters states with a distribution different from training.

We already note that the problem with length generalization is not necessarily related to having large norms in the state; rather, it is due to having states whose distribution change after the training context: the State Passing intervention (Section 4.1) achieves length generalization but has a higher standard deviation in the state at almost all positions (Figure 1). In Section C we show that this change in distribution is due to specific specific layers and heads of the state.

### 3.6. Unexplored States Hypothesis

Inspired by our previous empirical findings, in this subsection we arrive at a hypothesis to understand length general-

---
[2]Note that because total variation is bounded between 0 and 1, $0 \leq \text{EffRem}_T(t) \leq 1$.

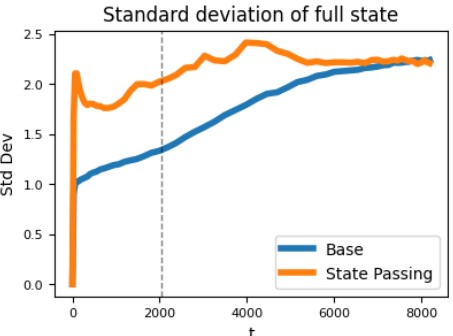

Figure 4. Standard deviation of the full state of the Mamba-2 130m official checkpoints versus the sequence position $t$ ($h_t$ in the notation of Section 2.2). The standard deviation is taken across all elements of the state in all layers. The Mamba-2 130m post-trained with State Passing (Section 4.1) produces states whose standard deviation do not significantly change after the training context $T = 2048$. In contrast, the official Mamba-2 checkpoint reaches a standard deviation almost twice as large at position $t = 8192$ than the one at position $t = 2048$.

ization based on the distribution of the state. At time $t$, recurrent models take an input $x_t$ and the recurrent state $h_t$ to output the prediction $y_t$. That is, we can write $y_t = f(x_t, h_t)$ for some function $f$. Therefore, the key to length generalization is understanding on which distributions of states and inputs the function $f$ has good performance.

For many tasks, we can assume that the distribution of inputs at time $t$ does not change much after the initial tokens (for example, in text, we expect that the marginal distribution of tokens at two different positions $t$ and $t'$ should not be too different). Thus, we can focus on the distribution of the state. In particular, we are interested in the distribution of *attainable states*, that is, the ones that the model would produce when processing a sequence.

Again, we denote by $\mathcal{D}_t := h_t(X_{0:t})$ the distribution of the state at time $t$. When a model is trained with a context length of $T$, it is exposed to state distributions $\mathcal{D}_t$ for $0 \leq t < T$, however it is never exposed to distributions $\mathcal{D}_{t'}$ for $t' \geq T$. If the distribution $\mathcal{D}_{t'}$ is different to the ones seen during training, the model is not guaranteed to have good performance. Based on this insight, we present our hypothesis for why models fail to length generalize:

***Unexplored states hypothesis***: *Recurrent models fail to length generalize when they are trained only on a subset of all attainable state distributions—i.e. on a subset of the states that would be attained if the state recurrence was rolled out indefinitely. When trained for long enough, the model overfits to this subset and performs poorly on long sequences because it encounters unexplored state distributions.*

This hypothesis explains the previous behavior we have observed. When the training context $T$ is short, the model overfits to a limited number of distributions $\mathcal{D}_t$ for $0 \le t < T$, which can be different to the distribution of the states attained after rolling out the recurrence on long sequences. Conversely, if the training context $T$ is increased, the model is trained on a wider range on distributions and is more likely to perform well on the distribution of attainable states.

# 4. Training Interventions to Enable Length Generalization

In the previous section, we verified that training with long sequences enables length generalization, which is in agreement with the *unexplored states hypothesis*. In this section, we further support this hypothesis by showing that length generalization is also achieved when the model is exposed to initial states that are similar to attainable states—in other words, training with long sequences is sufficient but not necessary for generalization. To do so, we propose four different interventions to initialize the state and study the generalization performance of recurrent models post-trained with those interventions.

## 4.1. SSM Equations For a Non-Zero Initial State

First of all, we will derive the equations for an SSM whose initial state is not zero initialized. Assuming $h_{-1} \ne 0$, we can unroll Equation 1a to have:

$$h_t = \sum_{s=0}^{t} A_{s+1:t}^{\times} B_s x_s + A_{0:t}^{\times} h_{-1} \tag{4a}$$

$$y_t = C_t^T \sum_{s=0}^{t} A_{s+1:t}^{\times} B_s x_s + C_t^T A_{0:t}^{\times} h_{-1} \tag{4b}$$

Thus, an SSM that is initialized with $h_{-1} \ne 0$ differs from a zero-initialized SSM only in additive factors of $A_{0:t}^{\times} h_{-1}$ and $C_t^T A_{0:t}^{\times} h_{-1}$ in the final state and output, respectively. As we mentioned in Section 2.2, most recurrent models can be formulated using Equation 1, and thus Equation 4 is valid for them. An explicit derivation of the formula is given in Section H.

## 4.2. Intervention 1: Random Noise

For the first intervention, we initialize all the values of the state by sampling independently from a Gaussian $\mathcal{N}(0, \sigma^2)$. We post-train the official Mamba-2 checkpoints for 100 steps with this technique ($\sim 0.02\%$ of the pre-training budget) and show the results in Figure 5 (dashed red line). This simple method slightly improves the generalization of the 370m model, but does not work for the 780m and 1.3b models. We believe the reason why this method does not work is that the distribution of attainable states of the Mamba-2

model is very different from an IID Gaussian with same mean and variance in all layers, especially for large model sizes. Thus, even though this non-zero initialization exposes the model to more state distributions, they are not realistic and do not cover the distribution of attainable states, which explains why it does not fix length generalization. More details on the post-training procedure are given in Section B.2.

An ablation on the impact of $\sigma$ is shown in Section F. We also note that the random initial state is applied only in training, while a zero-initialized state is used for validation. Introducing noise in validation significantly degrades the model's performance on the first tokens.

## 4.3. Intervention 2: Fitted Noise

For the second intervention, we initialize the state from a distribution that resembles attainable states more closely. To do so, we sample from a Gaussian distribution with mean and standard deviation fitted to the final states seen during training. More concretely, during training for each layer $l$ and head $h$ we dynamically compute the mean and variance of the final states using a moving average with $\beta = 0.1$:

$$\mu^{(li)} \leftarrow (1 - \beta)\text{Mean}(h_T^{(li)}) + \beta \mu^{(li)} \tag{5}$$

$$\sigma^{2(li)} \leftarrow (1 - \beta)\text{Variance}(h_T^{(li)}) + \beta \sigma^{2(li)} \tag{6}$$

where the Mean and Variance are taken across the head dimension and state expansion dimension $N$. Then, at each training step the values of the initial state of the layer $l$ and head $h$ are sampled independently from a Gaussian with the updated parameters $\mathcal{N}\left(\mu^{(li)}, \sigma^{2(li)}\right)$.

We apply this intervention by post-training the official Mamba-2 models for 100 steps ($\sim 0.02\%$ of the pre-training budget) and present the results in Figure 5 (dashed orange line). This approach significantly enhances the generalization of the 370m and 780m models but still fails to correct the 1.3b model. We hypothesize that this is because independent sampling does not generate realistic states for the 1.3b model: as the model size increases, the states likely become more complex, with stronger dependencies between values. As in the previous intervention, we note that Gaussian state initialization is used only during training.

## 4.4. Intervention 3: State Passing

In this intervention, we initialize the state to the final state of a different sequence (for convenience, we shuffle all the sequences and use the final states of the previous batch during training). Note that this is equivalent to sampling an initial state from the distribution of attainable states. Additionally, we want the model to perform well on sequences with no prior context. To achieve this, we implement a dropout

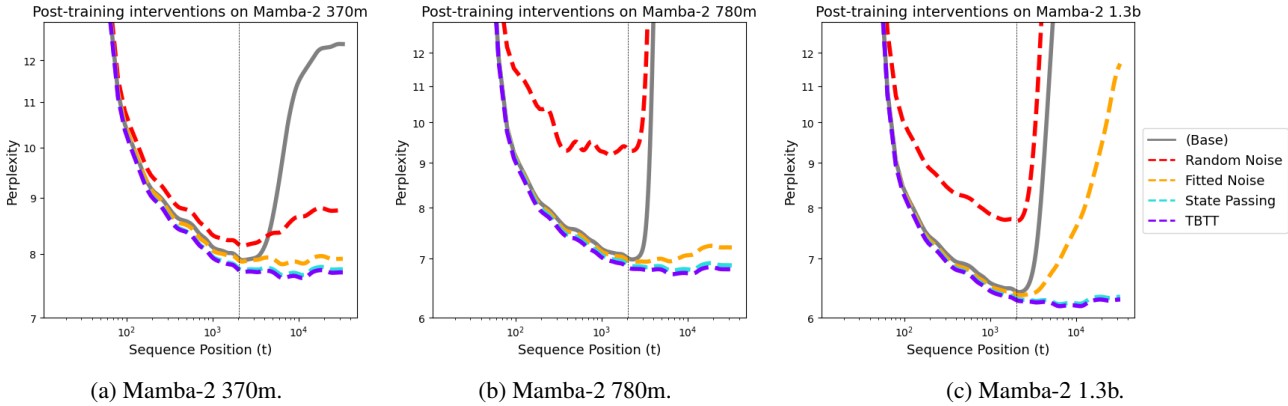

(a) Mamba-2 370m.           (b) Mamba-2 780m.           (c) Mamba-2 1.3b.

*Figure 5.* Position-wise perplexity of official Mamba-2 models (BASE) and our four interventions that are applied to these models with 100 post-training steps. For the STATE PASSING and TBTT interventions, 100 steps is enough to enable length generalization in 32k length sequences for all models. The interventions modify the initial state of the recurrent models, thus facilitating the exploration of a wider range of states. They sample an initial state from distributions that progressively get closer to the true distribution of attainable states: (1) RANDOM NOISE samples from a Gaussian distribution with fixed variance; (2) FITTED NOISE samples from a Gaussian distribution with mean and variance calibrated to the final states seen during training; (3) STATE PASSING uses the final state of a different sequence as initial state; and (4) TBTT splits a sequence into several chunks and uses the final state of the previous chunk as initial state. STATE PASSING directly samples from the distribution of attainable states and together with TBTT has the best performance, supporting the *unexplored states hypothesis*. For STATE PASSING, the results for Mamba-1 and Gated Linear Attention are also shown in Figure 1.

mechanism that randomly zero-initializes the state with a probability of $p = 0.1$.

The results for this intervention are shown in Figure 5 (dashed blue line). With only 100 steps ($\sim 0.02\%$ of the pre-training budget), this intervention enables length generalization, which supports our hypothesis that the failure to length generalize is due to the model not being trained on attainable state distributions.

In Figure 1, we also apply this techniques to enable length generalization in both Mamba-1 and GLA. Additionally, we show the results for Effective Remembrance in these intervened models. The Effective Remembrance values are smaller than the baseline, especially for $t \ll T$, suggesting that the intervened models are not substantially affected by far away tokens and correctly model the recent context. Interestingly, the Effective Remembrance curves for the Mamba models and GLA models have different shapes. In GLA, the tokens that are more than 512 positions away from $T$ have a small impact on the output, but the model is extremely affected by tokens that are just 512 positions apart (we recall that 512 is the training context length for the GLA models we are evaluating).

Additionally, in Figure 4 we show that a model post-trained with State Passing produces states whose distributions do not seem to significantly change over time, which sheds light into how this intervention achieves length generalization: the state update mechanism is modified so that the distribution of attainable states does not change much after the training context.

### 4.5. Intervention 4: Truncated Backpropagation Through Time (TBTT)

For the fourth intervention we use Truncated Backpropagation Through Time (TBTT) (Williams & Peng, 1990; Sutskever, 2013). This method consists in splitting a long sequence into smaller chunks, and for each chunk using the final state of the previous chunk as initial state.[3] Even though the gradient propagation is stopped between chunks, the model still learns to model a sequence based on some previous context (initial state).

A comparison with the rest of the interventions is shown in Figure 5 (purple dashed line). It is worth noting that this method achieves very similar performance to State Passing, thus indicating that they key for length generalization is exploring attainable state distributions, not necessarily processing (chunked) long sequences.

## 5. Performance of Interventions on Long Context Tasks

In the previous section we showed that the fitted noise, State Passing and TBTT interventions help length generalization on the Pile (Gao et al., 2020). As we mentioned in Section 3.4, length generalization would also be possible if the model only used the last $T$ tokens of a sequence to output

---

[3]From an implementation point of view, the difference with State Passing is that TBTT does not shuffle the sequences and requires carefully setting up the dataloader sampler so that the final state of a chunk is used as initial state for the next chunk.

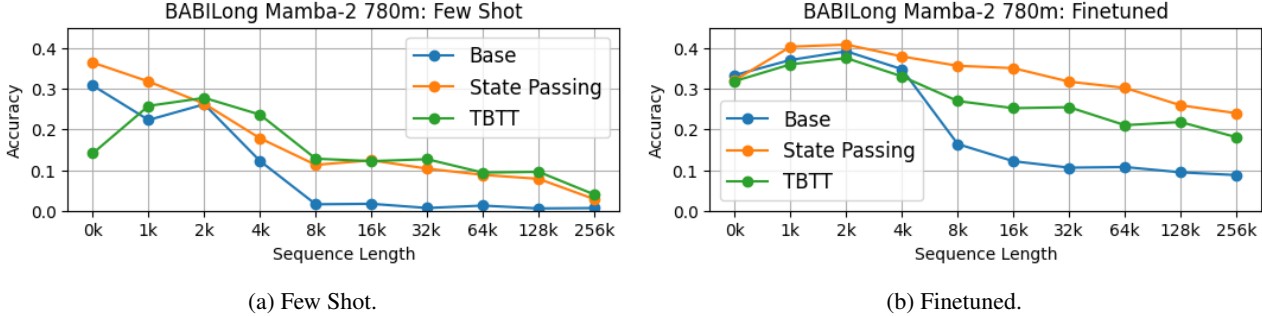

(a) Few Shot.  (b) Finetuned.

*Figure 6.* Few shot and finetuned performance for the BABILong benchmark (Kuratov et al., 2024) of a Mamba-2 780m model under three settings. BASE corresponds to the official checkpoint; STATE PASSING is the official checkpoint post-trained with State Passing on the Pile (see Section 4.1); and TBTT corresponds to the official checkpoint finetuned with Truncated Backpropagation Through Time on the Pile (see Section 4.5). The STATE PASSING and TBTT interventions significantly improve the baseline in both the few shots and finetuned settings, achieving reasonable performance in sequences up to length $256k$ despite only being trained with a context of $2k$. More finetuning details on Section B.3

.

its prediction (where $T$ is the training context). However, this is not a desirable behavior, as the model would fail in tasks that require reasoning over tokens that are more than $T$ positions apart. Additionally, length generalization was measured using perplexity, but no other metrics were explored. Thus, an important question remains: *Are these interventions truly effective in tasks that require reasoning over sequences longer than the training context $T$?* In this section, we answer affirmatively by showing results on three long context tasks.

### 5.1. Long Context Reasoning: BABILong benchmark

BABILong (Kuratov et al., 2024) is a challenging benchmark which tests both the common sense understanding of a model as well as its ability to capture long range dependencies in text. In Figure 6, it can be observed that State Passing enhances the length extrapolation capabilities of the model in both the few shot and finetuned settings (we recall that the model is trained and finetuned on sequences of length 2048). Therefore, State Passing is not only useful in fixing the diverging perplexity of established language models, but also in enhancing their ability to solve long context reasoning tasks.

### 5.2. Long Context Passkey Retrieval

The passkey retrieval task (Mohtashami & Jaggi, 2023) requires the model to retrieve a 5-digit passkey inserted at a given depth of a long context. In Figure 8 we present results for two Mamba-2 model sizes under three different settings: zero shot evaluation, standard finetuning with context length $T = 2048$ and finetuning with the fitted noise intervention, also with $T = 2048$ (see Section B.4 for why we use the fitted intervention in this task). The models finetuned with

fitted noise are capable of handling relationships between tokens that are much more than 2k positions apart. Moreover, they achieve better performance on much harder tasks; in particular the 780m achieves perfect accuracy on sequences of length 256k. We provide more details on the tasks and finetuning recipe on Section B.4.

### 5.3. Synthetic Copying

The synthetic copying task (Jelassi et al., 2024) consists in copying an arbitrary sequence of tokens. In Table 1 we present the results for a Mamba-2 model that is trained from scratch, and we show that using State Passing during training greatly improves length generalization in sequences more than three times longer. Thus, State Passing helps the model solve long context tasks that are harder than those seen during training.

| ARCHITECTURE | CHARACTER ACCURACY |
|---|---|
| Mamba-2 | $0.27 \pm 0.03$ |
| Mamba-2 + State Passing | $0.47 \pm 0.11$ |

*Table 1.* Length generalization results for the synthetic copying task (Jelassi et al., 2024), which consists of copying an arbitrary sequence of tokens. The models have 45 million parameters, are initialized randomly and trained on 1 million sequences of length between 50 and 100. They are evaluated on sequences of length 300. The State Passing intervention greatly improves length generalization.

## 6. Related Work

**Length generalization in Mamba through changes in the state update mechanism.** Some works have proposed changing the internal computation of Mamba, for exam-

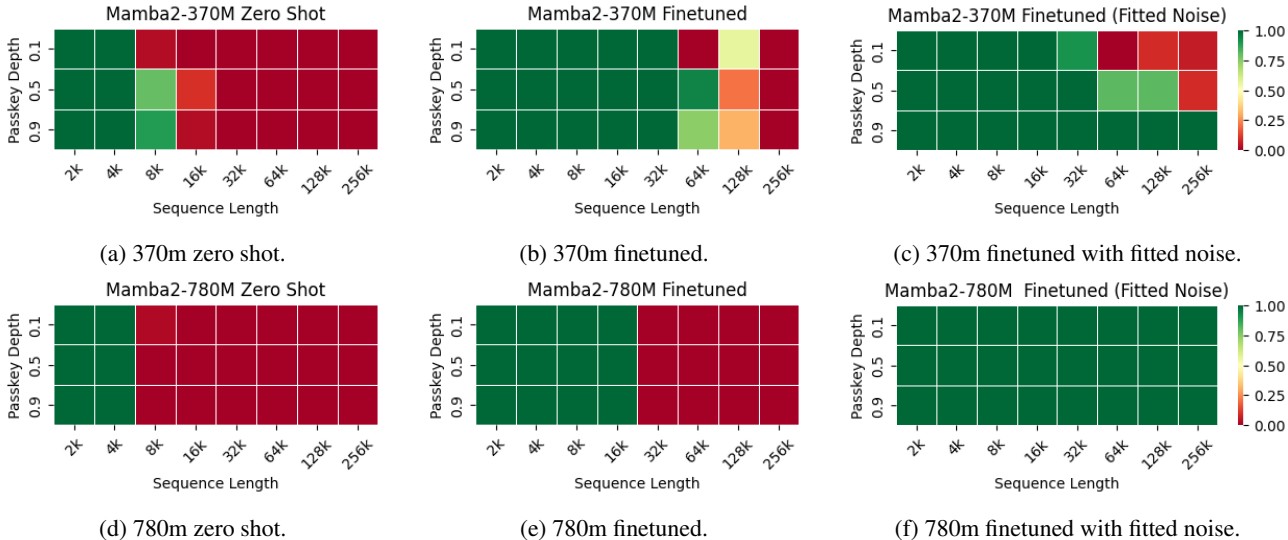

*Figure 7.* Performance on the passkey retrieval task (Mohtashami & Jaggi, 2023) of the official Mamba-2 checkpoints under three settings: (left) zero shot, (middle) standard finetuning, (right) finetuning with fitted noise. Finetuning with fitted noise enables solving the passkey task on sequences at least twice as long than with standard finetuning. Additionally, although the models are finetuned with sequences of length 2048, the fitted noise intervention solves tasks that require leveraging dependencies between tokens that are more than 2048 positions apart. More details on finetuning are given in Section B.4.

ple by updating the state of Mamba only on some selected tokens—which is equivalent to shortening the effective context (Ye et al., 2025; Ben-Kish et al., 2024; Yuan et al., 2024), or by forcing the model to forget previous context (Chen et al., 2024b). While these methods show increased performance in tasks like passkey retrieval and document retrieval (Mohtashami & Jaggi, 2023; Rajpurkar et al., 2018), a downside is that they change the internal state update mechanism and might be hard to transfer to architectures different to Mamba. In contrast, our interventions do not change the implementation of the architecture and are applicable to several recurrent models.

**Length generalization in SSMs and overfitting to short sequences.** Some other works link the failure to length generalize of SSMs with state overparametrization and overfitting to short sequences. Wang (2024) identifies that zero-initialized recurrent models struggle with length generalization, provides a theoretical analysis that links length generalization with polynomial extrapolation, and proposes pre-training models with State Passing and TBTT (Williams & Peng, 1990; Sutskever, 2013) to enable length generalization. Chen et al. (2024b) shows that training models for longer sequences enables length generalization and argues that the failure to length generalize is due to the model being overparametrized for its training context and not learning to forget past tokens. We build upon these insights to propose the *unexplored states hypothesis* as a precise explanation for why recurrent models fail to generalize. We focus on the distribution of the states and specify on which distributions the

model needs to be trained to enable length generalization—namely, distributions that are close to the distribution of final states—reaching the conclusion that training on long sequences or with TBTT is not always necessary.

## 7. Conclusion and Future Work

This work presents an empirical and theoretical study of the length generalization of recurrent models. We introduce the *unexplored states hypothesis*, which states that the failure to length generalize is due to the model not being trained on the distribution of the states that would be attained if its recurrence was applied to long sequences. Besides bringing important insights about the states and the behavior of recurrent models through tools like Effective Remembrance, this works also presents simple and general tools to enable length generalization for general recurrent architectures.

We mention several directions for future work. Firstly, while our work mostly focuses on length generalization in tasks related to text modeling, a further study of length extrapolation in other distributions with long range dependencies would be insightful. Secondly, we chose Mamba, Mamba-2, Gated Linear Attention and RWKV-v6 as a representative subset of modern recurrent architectures, but it would be interesting to analyze the training interventions on other recurrent architectures, for example on time invariant models like S4 (Gu et al., 2022) or hybrid models (Lieber et al., 2024; Ren et al., 2024; De et al., 2024; Arora et al., 2024)).

## Acknowledgements

RBR was supported by the "la Caixa" Foundation (ID 100010434). The fellowship code is LCF/BQ/EU22/11930090. We thank Aakash Lahoti for assistance with the synthetic copying experiment.

## Impact Statement

This paper presents work whose goal is to advance the field of Machine Learning. There are many potential societal consequences of our work, none which we feel must be specifically highlighted here.

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

*Table 2.* Model configurations and training hyperparameters for our experiments. The learning rates follow the values of previous works (Brown et al., 2020; Biderman et al., 2023).

| ARCHITECTURE | PARAMS | n_layers | d_model | n_heads / d_head | d_state | LEARNING RATE | BATCH SIZE |
|---|---|---|---|---|---|---|---|
| Mamba-1 | 130m | 24 | 768 | - | 16 | 6e-4 | 0.5M tokens |
|  | 350m | 48 | 1024 | - | 16 | 3e-4 | 0.5M tokens |
| Mamba-2 | 45m | 12 | 512 | 8 / 64 | 128 | 1e-3 | 0.5M tokens |
|  | 85m | 12 | 768 | 12 / 64 | 128 | 1e-3 | 0.5M tokens |
|  | 130m | 24 | 768 | 12 / 64 | 128 | 6e-4 | 0.5M tokens |
|  | 350m | 48 | 1024 | 16 / 64 | 128 | 3e-4 | 0.5M tokens |
|  | 780m | 48 | 1536 | 16 / 96 | 128 | 2.5e-4 | 0.5M tokens |
|  | 1.3b | 48 | 2048 | 32 / 64 | 128 | 2e-4 | 0.5M tokens |
| GLA | 70m | 6 | 512 | 4 / 128 | - | 1e-3 | 0.5M tokens |
|  | 120m | 6 | 768 | 4 / 182 | - | 6e-4 | 0.5M tokens |
|  | 360m | 20 | 1024 | 4 / 256 | - | 3e-4 | 0.5M tokens |

## A. Extended Related Work

**Length generalization in Transformers.** In order to length generalize, Transformers face the added difficulty of handling positional encoding, which is not easily extendable beyond the training context (Zhao et al., 2024). To deal with this issue, some works have proposed several position interpolation techniques (Chen et al., 2023b; Ding et al., 2024; Kazemnejad et al., 2023; Zhu et al., 2024; Peng et al., 2024b). Other works have proposed finetuning with sparse attention to allow more efficient length generalization (Chen et al., 2024a; Mohtashami & Jaggi, 2023), or they have developed methods to maintain a fixed-size sliding window for the KV cache (Xiao et al., 2024). These methods are bespoke to the usage and architecture and not always achieve good performance in long context tasks (Press et al., 2022; Li et al., 2024; Chen et al., 2023a). In contrast, recurrent models can in principle extend beyond their training context by simply rolling out the state recurrence. In this work, we show that there are simple and general interventions to enable length generalization in recurrent models, thus fully leveraging their benefits.

**Algorithmic extrapolation.** Some works have evaluated recurrent models on algorithmic tasks that are harder to solve than the ones the model has been trained on, and they have proposed training modifications that prevent the model from overfitting to the simpler training tasks (Bansal et al., 2022; Veerabadran et al., 2023). In particular, Bansal et al. (2022) proposes Incremental Progress Training, which reuses states from previous iterations of the task to discourage the model from learning iteration-specific behaviours, which is similar to why the TBTT intervention works in text. While there are some differences in the tasks (e.g. in these algorithmic extrapolation tasks the state should converge to a fix point), some of the techniques used are similar. This work presents the *unexplored states hypothesis* and a framework to reason about length generalization based on the distribution of states, which helps understand why these interventions work. We leave it as future work to study them in a broader set of cases.

## B. Training recipes

### B.1. Pre-training Language Models

For the experiments in Section 3, we train on the Pile (Gao et al., 2020) with the `EleutherAI/gpt-neox-20b`[4] tokenizer (Black et al., 2022). For the learning rate, we use cosine scheduling with warmup in the 10% first training steps, a peak learning rate given by Table 2 and a decay to $1e - 5$. The gradients are clipped to $1.0$ and no dropout is used. Additionally, we also follow the improved training recipe of Grattafiori et al. (2024), with an Adam optimizer with $\beta_1 = 0.9$ and $\beta_2 = 0.95$, weight decay scheduling with a peak of $0.01$, RMSNorm (Zhang & Sennrich, 2019) instead of LayerNorm and no linear biases. For Mamba-1 and Mamba-2 we use a training context of 2048 and for GLA we use a context of 512. Depending on the experiment, the models are trained for several times what Chinchilla scaling laws dictate (Hoffmann et al., 2024), see Figure 2 and Figure 3.

---

[4]https://huggingface.co/EleutherAI/gpt-neox-20b

### B.2. Post-training Interventions on the Pile

For the post-training interventions of Section 4, we use the same recipe as for model pre-training (section B.1) yet using a peak learning rate that is ten times smaller than the one given in Table 2. We post-train for 100 steps.

### B.3. Babilong Finetuning

The models are evaluated on the tasks [qa1, qa2, qa3, qa4, qa5, qa6, qa7, qa8, qa9, qa10] of the benchmark, and finetuned on such tasks for one epoch using facts and questions from the BABI training dataset (Weston et al., 2016). In the finetuned setting, all the models are finetuned on BABILong without State Passing nor TBTT, thus the benefits of having a State Passing or TBTT finetuned checkpoint are not lost when finetuning again for this task.

### B.4. Passkey Retrieval Finetuning

Contrary to typical language modeling datasets, the distribution of tokens in the passkey task is not stationary. This is why we show results for the fitted noise intervention, as it does not require using the final state of the sequence (i.e., right after revealing the passkey), which might not be appropriate as the initial state.

We finetune the official Mamba-2 checkpoints on the passkey retrieval task using the same procedure as section B.1, this time for 1000 steps and using a peak learning rate ten times smaller than the one given in Table 2. In order to finetune, we mask all the tokens in the sequence that do not correspond to the passkey, trim the filler sentence to have a uniform batch size, and sample a different passkey depth between 0 and 1 for each sample. Additionally, we add a period after the passkey (".") because we observed that some models failed because they repeated the passkey several times consecutively (i.e. for a passkey of 12345, the output contained 12345123451234512345...). An example of a sample from the dataset is shown in Figure 8.

---

There is an important info hidden inside a lot of irrelevant text. Find it and memorize them. I will quiz you about the important information there.
is green. The sky is blue. The sun is yellow. Here we go. There and back again. The grass is green. The sky is blue. The sun is yellow. Here we go. There and back again. The grass is green. The sky is blue. The sun is yellow. Here we go. There and back again. The grass is green. The sky is blue. The sun is yellow. Here we go. There and back again. The
The pass key is **3327**. Remember it. **3327** is the pass key.
The grass is green. The sky is blue. The sun is yellow. Here we go. There and back again. The grass is green. The sky is blue. The sun is yellow. Here we go. There and back again. The grass is green. The sky is blue. The sun is yellow. Here we go. There and back again. The grass is green. The sky is blue. The sun is yellow. Here we go. There and back again.
What is the pass key? The pass key is **3327**.

---

*Figure 8.* Sample of the passkey retrieval task (Mohtashami & Jaggi, 2023) with length 256 and depth 0.5.

## C. Distribution of the State at Given Heads and Layers Over Time

Figure 9 shows the standard deviation of certain heads of the Mamba-2 state as a function of time. In the official checkpoint, the standard deviations increase after the context length, thus producing a distribution shift which explains why the model fails to length generalize. In contrast, the State Passing intervention fixes this issue by producing states whose standard deviations are more stable after the training context.

## D. Position-wise Perplexity for RWKV-v6

Figure 10 shows the position-wise perplexity of a RWKV-v6 model (Peng et al., 2024a) trained from scratch with a context of $T = 256$ and post-trained with State Passing. When the training context is too short, the RWKV-v6 model fails to length generalize (as we discussed in Section 3.2), whereas the State Passing intervention (Section 4.1) achieves length generalization.

## E. Impact of the Warm-Up Period on Length Generalization

In Figure 11 we show the results of pre-training a Mamba-2 45m model from scratch with two different learning rate warm-up periods and a context length of $T = 1024$. Having an extended warm-up period hinders length generalization

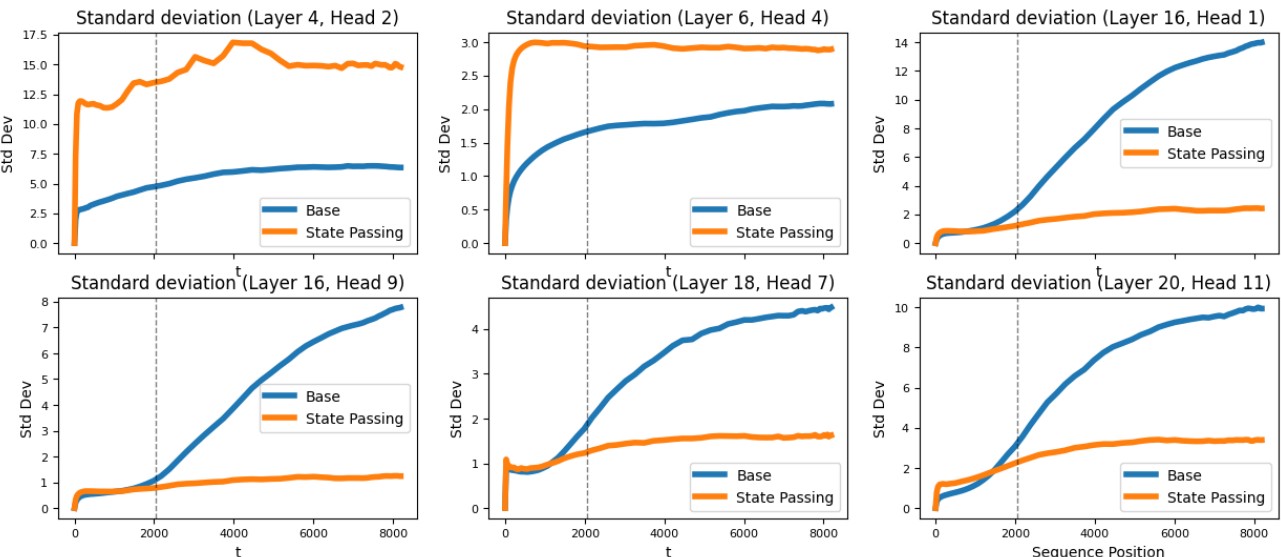

*Figure 9.* Standard deviation of parts of the state of the Mamba-2 130m official checkpoint versus the sequence position $t$ ($h_t$ in the notation of Section 2.2). As in the case of Figure 4, the model post-trained with State Passing produces states whose standard deviations are more stable than the baseline across time. The layers and heads are selected based on which had more variation across time, the majority of other heads in the baseline model do not exhibit this abnormal behavior. Thus, the distribution shift that is observed in Figure 4 is only due to specific heads in the state. Sometimes, the State Passing intervention generates states with a higher standard deviation than the pre-trained model (see Layer 4, Head 2 and Layer 6, Head 4). Thus, we note that the solution to length generalization is not avoiding large standard deviations in the state; but rather having states whose distribution do not significantly change after the training context.

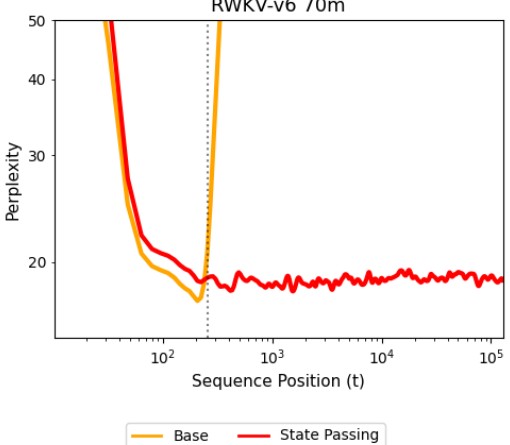

*Figure 10.* Position-wise perplexity of a RWKV-v6 70m model (Peng et al., 2024a) pre-trained from scratch for 35B tokens and post-trained with State Passing for 250m tokens on The Pile (Gao et al., 2020) (see B for details). It exhibits a similar behavior to the models shown in Figure 1: when pre-trained in a short context for many tokens it fails to length generalize, but post-training with a small amount of steps with State Passing enables length generalization.

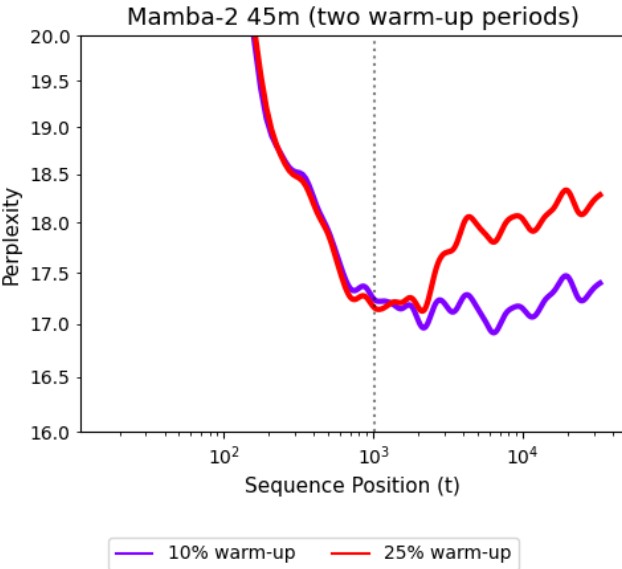

*Figure 11.* Mamba-2 45m trained on a context of $T = 1024$ with two different warm-up periods of the learning rate scheduler (10% and 25% refer to having a warm-up period that lasts 10% and 25% of the full training, respectively). An increased warm-up period hinders performance beyond the training context, demonstrating the impact of the training recipe for length generalization. The models are trained for 9B tokens (10x Chinchilla scaling laws) with the recipe given in Section B.1.

in this case, suggesting that length generalization can be affected by training hyperparameters. Our work removes the need to carefully analyze and tune these hyperparameters, since length generalization is expected to be achievable with the interventions we propose in Section 4.

## F. Impact of the Standard Deviation on the Random Gaussian Initial State Intervention

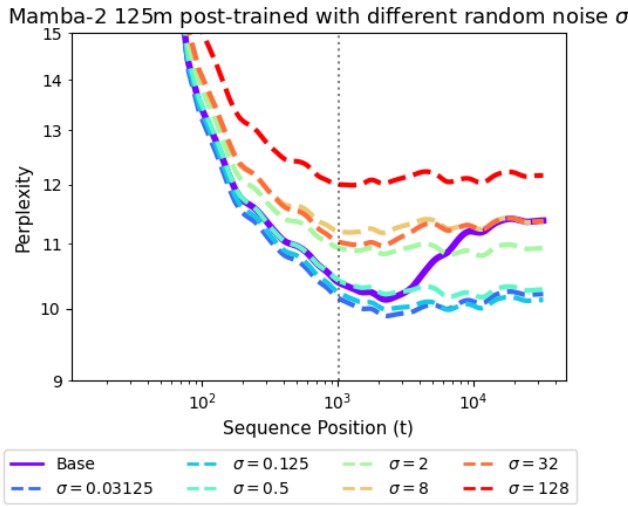

*Figure 12.* Position-wise perplexity of Mamba-2 125m post-trained for 100 steps with random initial state at different standard deviations $\sigma$. This post-training technique enables length generalization, but if the noise level is too high it hurts performance.

In Section 4.2, we proposed sampling the initial state from a Gaussian distribution of standard deviation $\sigma$ (see Section 4.2), which is a parameter that needs to be tuned for each model. In Figure 12 we show the position-wise cross entropy of a 125m Mamba-2 model post-trained with different standard deviations in the random initial state. This post-training technique

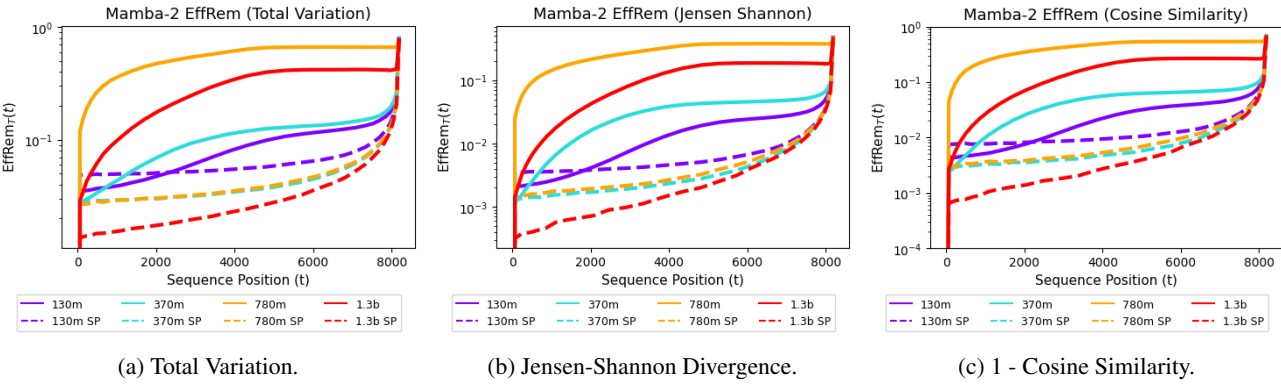

(a) Total Variation.     (b) Jensen-Shannon Divergence.     (c) 1 - Cosine Similarity.

*Figure 13.* Comparison of different distance metrics for the computation of Effective Remembrance (Section 3.4). The models shown are the Mamba-2 official checkpoints (solid lines), and their post-trained counterparts with State Passing (Section 4.1). They are evaluated on the valildation dataset of the Pile (Gao et al., 2020), and the Effective Remembrancce $T$ value is 8192. For these models, the shape of the Effective Remembrance is robust to the choice of distance.

facilitates length generalization; however, an excessively high noise level degrades performance. On the other hand, we have observed that if the noise standard deviation is too small, this intervention does not enable length generalization.

## G. Ablation on the Choice of Distance for Effective Remembrance

In Section 3.4 we introduced Effective Remembrance as a metric to understand the impact of certain parts of the context in the output of an autoregressive model. Since Effective Remembrance compares two different outputs of the autoregressive model, which are probabilities, we used total variation as a distance between them. In this subsection, we explore the impact of using a different distance (the Jensen-Shannon distance) and also using the cosine similarity between the probability vectors. The results for the Mamba-2 baseline models and the same models post-trained with State Passing is shown in Figure 13. Although the absolute values vary, it can be seen that the shape of the Effective Remembrance does not change much depending on the distance.

## H. Derivation of SSM Equations for Non-Zero Initial State

Taking equation 1a and unrolling it:

$$
\begin{aligned}
h_t &= A_t h_{t-1} + B_t x_t \\
&= A_t \left( A_{t-1} h_{t-2} + B_{t-1} x_{t-1} \right) + B_t x_t = A_{t-1} A_t h_{t-2} + B_{t-1} A_t x_{t-1} + B_t x_t \\
&= ... \\
&= A_{0:t}^\times h_{-1} + \sum_{s=0} A_{s+1:t} B_s x_s
\end{aligned}
$$

## I. State Passing Pytorch Pseudocode

In section 4.1 we provide the equations to compute the output and final state of a recurrent model when the initial state is not zero-initialized. In Figure 14 we provide pseudocode to compute the contribution of the initial state to the final state and output.

```python
import torch
import torch.nn.functional as F

def state_passing(X : torch.Tensor,
                  A : torch.Tensor,
                  B: torch.Tensor,
                  C: torch.Tensor,
                  state: torch.Tensor,
                  ) -> tuple[torch.Tensor, torch.Tensor]:
    """
    B: Batch size
    T: Sequence Dimension
    H: Hidden dimension
    N: State dimension

    Inputs:
        X: (B, H, T) - Input Sequence
        A: (B, H, N, T) - A (Equation 1a)
        B: (B, N, T) - B (Equation 1a)
        C: (B, N, T) - C (Equation 1b)
        state: (B, H, N) - Initial state
    Outputs:
        y: (B, H, T) -  Contribution of the initial state to the output
        final_state: (B, H, N)  - Final state
    """

    # Generate A_{0:T}, A_{1:T}, ..., A_{T:T}
    A_cs_rev = torch.cumsum(A.flip(-1), dim=-1).flip(-1)
    A_cs_rev = torch.exp(F.pad(A_cs_rev, (0,1)))  # (B H N T)
    final_state = torch.einsum("bht,bnt,bhnt->bhn", X, B, A_cs_rev[:, :, :, 1:])
    # Final state contribution from initial state
    final_state = final_state + state * A_cs_rev[:, :, :, 0]

    # Calculate contribution of initial state on outputs
    A_cs = torch.exp(torch.cumsum(A, dim=-1))  # (B H N T)
    y = torch.einsum("bhn,bhnt,bnt->bht", state, A_cs, C)
    return y, final_state
```

*Figure 14.* Pseudocode to compute the final state (Equation 4a) and contribution to the output from the initial state (second term of the right hand side of Equation 4b) for a recurrent model when the initial state is not zero-initialized. This code assumes that the matrix $A_t$ is diagonal.

