# OpenReview forum: "Understanding and Improving Length Generalization in Recurrent Models"
_ICML.cc/2025/Conference — ICML 2025 poster_

### Official Review · Reviewer_5H8P · 2025-03-13

**Overall Recommendation:** 3

**Summary:**

The paper investigates why recurrent models fail to generalize to sequence lengths beyond their training context and proposes methods to improve length generalization. They also propose a metric (Effective Remembrance) which basically captures the difference in a model's next token distribution at a point, when considering two different context lengths (starting at t=0 or t=some other index in the past >0).

**Claims And Evidence:**

Most claims are well-supported, particularly the core hypothesis that unexplored states lead to generalization failure and that state-based training interventions can fix the issue.

**Essential References Not Discussed:**

-

**Experimental Designs Or Analyses:**

-

**Methods And Evaluation Criteria:**

The paper presents comparisons using only the perplexity metric on The Pile dataset (so the task is next token prediction), and on a very synthetic long context task (passkey retrieval). It would be good to show the results on a long context task derived from real world data (maybe LongBench or even something simpler that requires 2 or 3 hop reasoning).

**Other Comments Or Suggestions:**

-

**Other Strengths And Weaknesses:**

I think the proposed approach of trying to understand train-time randomisations/methods to encourage length generalization makes sense and is useful. Presenting results on a long-context generalization benchmark that is not very toy or synthetic would significantly strengthen the paper.

**Questions For Authors:**

Am I correct in assuming that state passing is different form TBTT since the state value (say s) used to intialise a new rollout at training iteration i, for batch element say 0 - is not related in any way to previous states that might have been seen in the prior context that corresponds to batch element 0?
So if we have a context of length 2N, but we are only able to train for context lengths of N, it is not the case that we are using the states at the end of N to initialize the rollout when we go from N+1 to 2N in the next batch (because the states at the end of the batch are shuffled - so while in some cases they might actually correspond to the correct context/trajectory - it many cases it will not?).

**Relation To Broader Scientific Literature:**

The paper shows that initialising the state of a sequence model with noise that has been fit to the typical distribution of how the states are at the end of a context window (say N), helps to train/finetune models with context length N, but genralise to 2N (since now the model has been trained while seeing what state distributions sort of look like at the end of 2N steps of context even if the gradient was not back propagated 2N steps). This tries to mimic what was effectively done when other works trained with TBTT and carried forward the state into a new rollout starting from the end point of the previous rollout of a sequence model - but this time with fitted noise or state passing - fitted noise intervention works well for the small models but not a bigger model, state passing works well for both.

**Theoretical Claims:**

No.

---

> ### Author Rebuttal · Authors · 2025-04-01
>
> We are encouraged to see that the reviewer thinks that our unexplored states hypothesis is well supported and that our interventions on the initial state are useful to achieve length generalization. We provide responses to their questions:
>
> > RE: Results on long-context generalization benchmarks besides passkey retrieval to strengthen the paper
>
> We followed the suggestion of the reviewer and **present new results for two long context benchmarks.**
>
> The first one is BABILong [1], a challenging benchmark which tests both the common sense understanding of the models as well as its ability to capture long range dependencies in text. We present results for the baseline and our interventions in the following link:
>
> https://postimg.cc/6TjDhBZ0
>
> **It can be observed that state passing enhances the length extrapolation capabilities of the model, improving performance from ~9% to ~24% in sequences of length $256k$** (we recall that the model is trained and finetuned on sequences of length $2K$). This reinforces our claim that state passing is not only useful to fix the diverging perplexity of established language models, but also improves helps length generalization by exposing them to more initial states during training and thus enhancing their ability to handle long contexts.
>
> The second benchmark is Synthetic Copying [2], which consists in copying an arbitrary sequence of tokens. The results are shown in the following link:
>
> https://postimg.cc/ctwryLQV
>
> **It can be seen that state passing greatly helps length extrapolation, improving the accuracy in sequences three times longer than those seen during training from 27% to 47%.** We believe these new experiments highlight the length extrapolation capabilities of the interventions on benchmarks that are closer to real world data, as the reviewer suggested.
>
> Finally, regarding the Longbench benchmark specifically, unfortunately we lack the computational resources to train and evaluate models that can achieve decent performance on this benchmark. Even models of more than 7B parameters are barely better than random guessing (see Table 2 in [3]), and we have focused our work on models in the 1B scale and below. However, we find this benchmark very relevant to our methods and will include it as an avenue for future work in the final version of the paper.
>
> [1] Kuratov, Y. et al.(2024). _BABILong: Testing the limits of LLMs with long context reasoning-in-a-haystack_. In _The Thirty-eighth Conference on Neural Information Processing Systems Datasets and Benchmarks Track_.
>
> [2] Jelassi, S. et al. (2024). Repeat After Me: Transformers are Better than State Space Models at Copying. <i>Proceedings of the 41st International Conference on Machine Learning</i>
>
> [2] Bai, Y. et al. (2024). LongBench: A bilingual, multitask benchmark for long context understanding. In Proceedings of the 62nd Annual Meeting of the Association for Computational Linguistics. Association for Computational Linguistics.
>
> > RE: The difference between state passing and TBTT resides in whether the initial state is related to the context of the current sequence
>
> That is correct: the difference between state passing and TBTT is that in state passing the initial state is unrelated to the given sequence, whereas in TBTT the initial state corresponds to prior context of the sequence. From an implementation point of view, the only difference is that in state passing the dataset is shuffled, whereas in TBTT the dataloader carefully orders the samples so that when passing a final state to the next sequence, such final state corresponds to the prior context of the sequence. As the reviewer points out, in principle it could be possible that in state passing the final state would correspond to the correct context, but the probability negligible given that the datasets are so large.
>
> Even though in practice the implementation differences are small, in theory the methods are doing very different things. Both methods can be seen as sampling initial states from a given distribution. State passing samples initial states from the distribution of final states (for convenience, we simply take the final state of the previous batch, which is a good enough approximation because the batches are unrelated). In contrast, TBTT samples states from a degenerate distribution, which specifically consists of the final state of the previous context of the sequence. A very interesting result of our paper is that TBTT is sufficient but not necessary for length generalization, given that state passing also works (or even sampling from random or fitted noise can be good enough). Besides being a practical and efficient way of enabling length generalization, we believe our results for these interventions shed light on the core issue with length generalization: models fail to length generalize when they are not trained on the distribution of states attained when processing a sequence (our "unexplored states hypothesis").

---

### Official Review · Reviewer_b5dM · 2025-03-14

**Overall Recommendation:** 3

**Summary:**

The authors study length generalization in recurrent models. They begin with an empirical analysis of length generalization failures for Mamba v1 / v2 and gated linear attention. They define a new metric, "Effective Remembrance", to quantify the influence of the context prefix on a model's predictions, and show that Mamba v2 has high Effective Remembrance for early tokens. They hypothesize that length generalization failures are due to incomplete exploration of the recurrent state distribution during training and propose four state initializations to mitigate this. They provide empirical results showing that post-training using these initialization schemes substantially improves length generalization measured in terms of perplexity.

**Claims And Evidence:**

The authors claim that:
1. Models fail to generalize when they reach states that were not explored during training (the "unexplored state hypothesis")
2. Length generalization in recurrent models can easily be improved by post-training with simple interventions on the state initialization.

The evidence for 2 is quite good given their experimental results, as long as you're comfortable characterizing length generalization in terms of perplexity. The evidence for 1 does not seem as strong. There is no direct investigation of the unexplored state hypothesis, even though this would seem to be straightforward - e.g by comparing the distribution of states under short vs long contexts and correlating their discrepancy with some measure of performance.

**Essential References Not Discussed:**

There is an active debate on whether perplexity is informative for long-context performance. Recent works include [1, 2], both of which observe low correlation between perplexity and performance on benchmarks such as LongBench [3]. It would be useful to cite this discussion and to hear the authors' thoughts given their approach in this paper.

[1] Fang, L., et al. (2024). What is Wrong with Perplexity for Long-context Language Modeling?. arXiv preprint arXiv:2410.23771.

[2] Hu, et al. (2024). Can Perplexity Reflect Large Language Model's Ability in Long Text Understanding?. arXiv preprint arXiv:2405.06105.

[3] Bai, Y., et al. (2023). Longbench: A bilingual, multitask benchmark for long context understanding. arXiv preprint arXiv:2308.14508.

**Experimental Designs Or Analyses:**

The experiments are sound overall. It might have been useful to include a "negative control" for the post-training, perhaps by post-training on the same amount of data with the standard zero-initialization for the state.

**Methods And Evaluation Criteria:**

Effective remembrance is an intuitive way to measure the influence of a context prefix. The interventions described for the state are clear, though there is not much effort to distinguish why you'd prefer one over the other, apart from the empirical results.

In terms of evaluation, length generalization is defined and evaluated almost entirely in terms of perplexity. While this is standard for the literature, it's worth noting that there meaningful debate as to whether perplexity is really informative for long-context performance (see below). The authors also study performance on a passkey task, but there are many other instances of long-context tasks, or even whole benchmarks, that could provide an alternative set of evaluation criteria for length generalization.

**Other Comments Or Suggestions:**

Typos:
- $x_t$, not $x_T$, in line 114?
- I'd suggest using "et al." for the 1.5-page Llama-3 citation

**Other Strengths And Weaknesses:**

The paper is well written and easy to follow. The claims are stated clearly and the methods are well defined.

**Questions For Authors:**

- Could the authors share their thoughts on using perplexity to evaluate length generalization?
- Why not test the unexplored state hypothesis directly by comparing state distributions in short and long context scenarios?
- The authors' effective remembrance analysis essentially concludes that the recurrent models (and Mamba v2 in particular) are too strongly influenced by observations in the distant past. This seems at odds to some extent with the main objective of modern recurrent architecture development, which is to produce models that are able to retain information over very long sequences. Moreover, it would seem that the "right" level of effective remembrance depends on the data generating process, so that less is not always better. Is there a balance to be struck here?

**Relation To Broader Scientific Literature:**

Length generalization is a long-running topic in sequence modeling, including for recent architectures like Mamba. Recurrent models have long been studied in terms of the statistical properties or dynamics of the state, including aspects that may lead to better performance over long sequences ("long memory") or more stable behavior.

**Theoretical Claims:**

N/A

---

> ### Author Rebuttal · Authors · 2025-04-01
>
> We thank the reviewer for its thoughtful analysis of the paper and concrete suggestions, and we are encouraged to see that they found the experiments sounds and the paper easy to follow. We provide responses to their discussion:
>
> > RE: Perplexity as an evaluation metric
>
> We understand the reviewer's concern that perplexity on its own might not be sufficient to asses the capabilities of the models. Thus, we have run two additional evaluations.
>
> (1) **BABILong** [1] : BABILong is a benchmark that tests the ability of the model to process a long context and respond to a natural language understanding question based on some facts in the context. The results are shown in the following link:
>
> https://postimg.cc/6TjDhBZ0
>
> **It can be seen that TBTT and especially state passing greatly help the model achieve a better performance than the baseline, improving from ~9% to ~24% in a context length of $256k$**.
>
> (2) **Synthetic copying** [2] : We have also evaluated length generalization on the synthetic copying task, which consists in copying an arbitrary sequence of tokens. The results are shown in the following link:
>
> https://postimg.cc/ctwryLQV
>
> **It can be observed that state passing greatly helps length extrapolation, improving the accuracy in sequences three times longer that those seen during training from 27% to 47%.** We believe these new experiments highlight the length extrapolation capabilities of the interventions on the initial state and hope they help address the reviewer's concern on perplexity as an evaluation metric.
>
> [1] Kuratov, Y. et al.(2024). _BABILong: Testing the limits of LLMs with long context reasoning-in-a-haystack_. In _The Thirty-eighth Conference on Neural Information Processing Systems Datasets and Benchmarks Track_.
>
> [2] Jelassi, S. et al. (2024). Repeat After Me: Transformers are Better than State Space Models at Copying. <i>Proceedings of the 41st International Conference on Machine Learning</i>
>
>
> > RE: Supporting the unexplored states hypothesis by comparing state distributions in short and long contexts
>
> We thank the reviewer for this suggestion, which we have added to the final version of the paper. **We measured the distribution of the state depending on the sequence position, and found that the standard deviation (across all elements of the state) increases as the sequence position increases. Thus, when processing longer sequences the model encounter states from a distribution that it has not seen during training. The results are in the following link:**
>
> https://postimg.cc/HcpL3yWP
>
> **Moreover, in the figure we show that our state passing intervention fixes this issue by producing states whose distributions do not vary so much based on the sequence position.** This effect can also be observed in specific heads of layers:
>
> https://postimg.cc/LJ0vZsZd
>
> This shift in distribution is correlated with performance: the models' perplexity diverges after the training context because it encounters a state that has not been seen during training (see Figure 1a). **Thus, we believe these new findings reinforce our unexplored states hypothesis.**
>
> > RE: Is lower Effective Remembrance always better?
>
> Yes, we absolutely agree that the "right" level of Effective Remembrance is not necessarily as low as possible. A certain level of Rffective Temembrance is needed for the model to remember tokens from the past (if it were zero at time $t$, it would be "effectively" ignoring the tokens in positions $[0,t]$, which is undesirable when long range dependencies are needed). However, the Rffective Remembrance curves of the baseline models shown in Figure 1 are clearly wrong, given that they are too influenced for tokens that are far away in the past. The state passing intervention yields more reasonable Effective Remembrance curves, where the model places more focus on the recent context, which is correlated with better performance.
>
> More generally, we believe Effective Remembrance can be a useful tool when applied to other settings and we are eager to use it in future work. Several recent works point out that linear recurrent architectures may fail to achieve tasks like associate recall or copying because they cannot remember past tokens or store previous context into the state (compared to transformers, which do not compress the past context). This could be verified by showing that Effective Remembrance is too low in these tasks (which would be undesirable here, as the reviewer suggested). However, in the context of our work (language models failing to length generalize), the opposite is actually the case: Effective Remembrance is too large. We hypothesize this is due to the states being overparametrized and due to the models overfitting to the states that arise when processing short sequences (which motivate our unexplored state hypothesis and the the use of non-zero initial states to train on a wider distribution of states and fix the issue).

---

### Official Review · Reviewer_8LQW · 2025-03-15

**Overall Recommendation:** 4

**Summary:**

The paper explores the reasons for limited length generalization of recurrent (mainly modern linear ones like mamba, GLA) neural networks.
The paper explores the hypothesis that this is due to "unexplored states" - i.e. the kind of states that occur after long context tend to be unfamiliar to models trained on shorter context (as such states are not attainable). The authors perform several empirical investigations surrounding this. The paper shows that the "effective remembrance" (a proposed measure that quantifies the influence of earlier tokens) is high for RNNs - and models struggling towards length generalization are highly affected by distant tokens. It shows training on short contexts can impede generalization further. All these suggests that the models struggle due to distribution shift of states that it encounter for longer sequence.
To counteract it the paper explores various post-training interventions:

1. Random Gaussian noise to initialize states  to increase the diversity of states a model encounters.
2. Fitted noise - from mean and variance of states occurring in longer contexts.
3. State passing - Passing some state of a different (randomly chosen) sequence end point as an initial state for another sequence.
4. Truncated Backpropagation Through Time.

The paper shows that random Gaussian and Fitted noise does enhance generalization to an extent for certain models but still limited because they don't necessary simulate realistic long-sequence states.

However, state passing and truncated backpropagation do enable effective generalization.

Later the paper also demonstrates maintenance of long-term dependence with Fitted noise for passkey retrieval task.

## Update after rebuttal

The authors address most of my concerns. I increased the score accordingly.

**Claims And Evidence:**

Mostly the claims are supported.

Few points I am a bit critical of:

1. The passkey retrieval task seems to be shown only for Fitted noise - which did not appear as an effective solution in Pile. So I would be curious:
     1. How does other methods particularly BPTT and state passing do on the task? If we are passing the state of different passkey-retrieval context - wouldn't that confuse the model?
     1. Does passkey retrieval also involve length generalization tests? If not, it's also desirable to see if the models can generalize in situations where actual long range dependency exists and is enforced (unlike general language modeling).

The paper also uses the general term "recurrent" to scope out the kind of models it explores instead of a more specific term like "linear recurrent". It doesn't seem to consider non-linear recurrent models (which have modern variants as well such as xLSTM [1]). Could be good to limit the scope more explicitly by specifying this if that's the intention.


[1] xLSTM: Extended Long Short-Term Memory - Beck et al. Neurips 2024

**Essential References Not Discussed:**

Not as familiar with this specific direction of length generalization in linear RNN.

However, this paper:

The Illusion of State in State-Space Models - Merill et al. ICML 2024

Seems relevant to a degree. This may suggest some root issues in SSMs - in state tracking; and that could also limit length generalization in certain contexts. Simply hacking the initial state may not be the resolution in this case.

xLSTM paper suggests better state-tracking - and should count as a model to further investigate if the title and abstract is kept general - i.e. making claims about "recurrent" neural networks and not "linear recurrent" neural networks more specifically.

**Experimental Designs Or Analyses:**

No strong issues stand out to me besides the one I already discussed.

One issue I may mention (which is broader than this paper - so I will not particularly penalize this paper for this) is as follows:

One point is that the context of length generalization matters. Simply length generalization in language modeling may not give the full picture because in the dominant distribution of natural language contexts -- it could be very well possible long-range dependencies and state tracking is not required. Even some popular solutions like AliBi and such seem to work by having more of a "forgetting" bias towards the past.

It's unclear, how well length generalization can be achieved in other contexts where the final result is very sensitive to simple changes in the past. Passkey retrieval can be one example, but there are other contexts as well one could explore like logical inference and ListOps length generalization - examples of such exploration include [1,2,3,4] among others.

Earlier SSMs like S4D were shown poor generalization there [4].

[1] Modeling Hierarchical Structures with Continuous Recursive Neural Networks - Ray Chowdhury et al. ICML 2021
[2] Ordered Memory - Shen et al. NeurIPS 2019
[3] The Neural Data Router: Adaptive Control Flow in Transformers Improves Systematic Generalization - Csordas et al. ICLR 2022
[4] Recursion in Recursion: Two-Level Nested Recursion for Length Generalization with Scalability - Ray Chowdhury et al. NeurIPS 2023

**Methods And Evaluation Criteria:**

Yes. Some potential misses are discussed above.

**Other Comments Or Suggestions:**

n/a

**Other Strengths And Weaknesses:**

**Strengths**

Overall, the paper is well-written. Despite any limitations it is a decent exploration of an idea. Exploration of the different post-training intervention techniques are interesting and can work as "quick fixes" for better generalization in the at least some contexts in the interim.


**Weakness**

1. Besides the things mentioned above, to an extent the hypothesis also feels a bit "obvious". I am not sure how much scientific weight to give to it.
Since recurrent models have the same parameters at every time step, in a sense, obviously, it should obviously generalize if the states are similar. And the corollary of that would be that if generalization is failing - the issue would be overfitting in the distribution of seen states.

1. It's not clear if we are getting an effective single intervention strategy that works both in general language modeling, and also in length generalization in contexts with long-range dependencies - passkey, listops etc. (part of which - especially for tasks requiring state-tracking - this might be also simply impossible for linear RNNs - given Merill et al's work cited above)

**Questions For Authors:**

1. Does passkey retrieval involve length generalization tests?
1. Are the other strategies BPTT/State-passing tested on passkey retrieval?
1. How about the zero-shot performance of the post-training intervention models on passkey?

**Relation To Broader Scientific Literature:**

The paper is connected to the lineage of linear recurrent models like RWKV, GLA, Mamba. Prior papers on linear recurrent models have already attempted different investigations regarding length generalization. This papers continues that direction but focusing more on the unexplored state hypothesis and non-architectural interventions - like BPTT and passing states from other models.

Key contributions seem to be theoretical formulation of a hypothesis why RNNs struggle to generalize and some initial empirical investigations both to support the hypothesis and explore some initial countermeasures to enable length generation.

**Theoretical Claims:**

As far as I have checked, no fundamental issues.

---

> ### Author Rebuttal · Authors · 2025-04-01
>
> We thank the reviewer for their detailed response and we appreciate that they find most our claims well supported, including our analysis on how state passing and TBTT enable length generalization because they simulate realistic states. We provide answers to their questions and observations:
>
> > RE: Does passkey involve length generalization?
>
> Yes, passkey retrieval involves length generalization, as the models have only been finetuned on samples of length $T=2048$, yet solve the task at length up to $256k$.
>
> > RE: Effectiveness of state passing when finetuning in passkey, and zero shot performance in passkey of models post-trained with interventions
>
> Originally, we decided to run the experiment with the fitted noise intervention, as in our opinion it is the more suited to this setting. As the reviewer mentions, it is somewhat unnatural to do state passing in this task, as it would be equivalent to providing a context containing a previous passkey. But we recognize that it is natural to wonder about the performance of state passing and TBTT, so here we also provide additional results for these two settings, where **overall state passing and TBTT bring some benefits to length generalization:**
>
> State passing finetuning in passkey: https://postimg.cc/vxwhrDXB
>
> Zero shot of TBTT post-trained checkpoints: https://postimg.cc/rKkcmJG6
>
> > RE: Do the interventions achieve length generalization for other long context tasks?
>
> We agree with the reviewer that perplexity and passkey retrieval might not be enough to assess the long context capabilities of the models, **so we have run experiments on two more long context tasks that are related to core aspects of language modeling**. Due to limited space, we refer to the response to Reviewer b5dM for more details on the tasks.
>
> Results for BABILong: https://postimg.cc/6TjDhBZ0
>
> Results for the synthetic copying task: https://postimg.cc/ctwryLQV
>
> **It can be seen that state passing and TBTT bring significant benefits in length extrapolation for these tasks**, for example by improving length extrapolation on sequences up to length 256k in BABILong. We also thank the reviewer for providing more related references on length generalization, which we will include in the paper as related work and avenues for future work.
>
> > RE: Scope of term "recurrent models"
>
> **We have added new results for another architecture, RWKV-v6, and we have observed the same phenomenon where the models trained on short context with zero-initialized states diverge, whereas post-training with state passing enables length generalization**
>
> https://postimg.cc/zLkhvF2x
>
> We hypothesize that other modern recurrent models will exhibit a similar behavior, but we agree with the author that all the studied models fall under the category of linear recurrent architectures that accept a formulation in terms of SSMs. Thus, we will change the wording in the final version to more clearly set the scope of the claims to the linear recurrent models for which we show experiments, and cite xLSTM as a different recurrent model which might have other behavior.
>
> > RE: the "unexplored state hypothesis" being obvious
>
> Indeed we agree with the reviewer that the hypothesis feels very intuitive in hindsight, but we believe our work is very relevant to the community for two reasons.
>
> Firstly, many recent works deal with length generalization by intervening on the architecture mechanism (see specific examples in Section 6), or by proposing new architectures and comparing their performance beyond the training context (for example see [1] or [2]). In contrast, our work: **(1) proposes the "unexplored states hypothesis" as a framework to understand the length generalization of recurrent models by reasoning about the distribution of states, and not the architecture mechanism**; **(2) systematically analyzes four training interventions which are motivated by the "unexplored states hypothesis**"; and **(3) shows that the interventions enable length generalization without changes to the architecture.**
>
> Secondly, even though it is natural to think that models fail to length generalize because they are out of distribution somehow, **we believe that it is a valuable contribution to propose training with several types of non-zero initialized states to fix this issue**. More specifically, we show that length generalization is enabled when the model is trained with initial states that resemble the states that are attained when processing long sequences. **We believe this finding is both intuitive and interesting, which we consider a strength of our work and hope makes a shift in the community towards simple (yet underutilized) training interventions with non-zero initialized states**.
>
>
> [1] Yang, S. et al. (2025). _Gated Delta Networks: Improving Mamba2 with Delta Rule_. In _The Thirteenth International Conference on Learning Representations_.
>
> [2] Sun, Y. et al (2024). _Learning to (Learn at Test Time): RNNs with Expressive Hidden States

---

> > ### Comment · Reviewer_8LQW · 2025-04-02
> >
> > Thank you for the rebuttal. It addresses some of my concerns and the additional results should strengthen the paper.
> > I increased the score to 4.

---

### Official Review · Reviewer_jMRg · 2025-03-16

**Overall Recommendation:** 2

**Summary:**

The authors in this paper propose a framework to analyze the problem of length generalization in recurrent networks. The authors primarily focus on State Space Models and how they behave when test sequences are significantly longer than training sequences by studying their response to 4 training interventions: (1) initializing the SSM with a random initial state, (2) learning the distribution from which the initial state from (1) is sampled, (3) initializing the SSM with the final state of a different sequence and (4) using truncated back propagation through time. The authors claim that the proposed simple interventions lead to significant improvement in length generalization when compared to vanilla Mamba SSMs trained without these interventions.

**Claims And Evidence:**

- I believe that the claims about state space models benefitting from the abovementioned training interventions is validated with sound experimentation.
- The authors however claim that they have developed a theoretical and empirical framework to understand length generalization broadly in recurrent models. This claim is not validated as the class of recurrent architectures evaluated in this work are (mostly just) SSMs and in addition, a specific type of hardware-efficient transformer (GLA).

**Essential References Not Discussed:**

I believe the two main interventions studied here (state passing and T-BPTT) are studied in other RNN models in prior art. So I don't agree that these interventions are novel contributions in this work. I believe that the authors have applied these interventions found in prior RNN length generalization literature on SSMs and GLA, which I believe is the contribution of the current work.

Prior findings list:
1. State passing: This is essentially what is studied under the name of Incremental Progress Training in the following paper on recurrent architectures.
Bansal, A., Schwarzschild, A., Borgnia, E., Emam, Z., Huang, F., Goldblum, M., & Goldstein, T. (2022). End-to-end algorithm synthesis with recurrent networks: Extrapolation without overthinking. Advances in Neural Information Processing Systems, 35, 20232-20242.
2. T-BPTT: Truncated back propagation through time is a well-known prior approach to learning stable recurrent models without gradient collapse as the authors have included in their references. This also is not a novel contribution of this paper.
3. There is a line of recent works in length generalization in recurrent models that is not mentioned in the related work. I request the authors to please look at this list of prior work in length generalization in RNNs, and add comment on how their proposed work is related to these highly related prior art.
i) Bansal, A., Schwarzschild, A., Borgnia, E., Emam, Z., Huang, F., Goldblum, M., & Goldstein, T. (2022). End-to-end algorithm synthesis with recurrent networks: Extrapolation without overthinking. Advances in Neural Information Processing Systems, 35, 20232-20242.
ii) Veerabadran, V., Ravishankar, S., Tang, Y., Raina, R., & de Sa, V. (2023). Adaptive recurrent vision performs zero-shot computation scaling to unseen difficulty levels. Advances in Neural Information Processing Systems, 36, 18132-18145.
iii) Goetschalckx, L., Govindarajan, L. N., Karkada Ashok, A., Ahuja, A., Sheinberg, D., & Serre, T. (2023). Computing a human-like reaction time metric from stable recurrent vision models. Advances in neural information processing systems, 36, 14338-14365.

**Experimental Designs Or Analyses:**

The authors have evaluated the effect of different training context lengths, number of trainable parameters and post-training interventions on a series of SSM architectures (Mamba-1, Mamba-2) and a hardware-efficient transformer architecture (GLA) in the context of length generalization. This experimental design does not necessarily reflect any lack of rigor, but like I mentioned, I find the task of passkey-retrieval to be quite simplistic and the choice of architectures to be incoherent with the overall phrasing of the contributions applying generally to all types of RNNs.

**Methods And Evaluation Criteria:**

- I believe the proposed methods and evaluation criteria are necessary and make sense for the problem of length generalization. However, the tasks used in this paper seem quite simplistic in nature and don't stress test the training interventions on more challenging length generalization problems.
- Using perplexity to measure length generalization I believe only measures the stability of the recurrent state space at test sequence lengths beyond what was experimented with during training. However, it doesn't study whether the extrapolation state space retains the same high expressivity (seen during training) in this unseen domain of longer recurrent sequences.
- I appreciate the authors evaluating their interventions on the passkey retrieval task, however, this task seems quite simplistic and would request the authors to consider tasks such as Maze Solving task in [1, 2] and the incremental grouping task in [3]. Evaluating on these more challenging tasks makes the work more impactful and practical.

References:
1. Bansal, A., Schwarzschild, A., Borgnia, E., Emam, Z., Huang, F., Goldblum, M., & Goldstein, T. (2022). End-to-end algorithm synthesis with recurrent networks: Extrapolation without overthinking. Advances in Neural Information Processing Systems, 35, 20232-20242.
2. Veerabadran, V., Ravishankar, S., Tang, Y., Raina, R., & de Sa, V. (2023). Adaptive recurrent vision performs zero-shot computation scaling to unseen difficulty levels. Advances in Neural Information Processing Systems, 36, 18132-18145.
3. Goetschalckx, L., Govindarajan, L. N., Karkada Ashok, A., Ahuja, A., Sheinberg, D., & Serre, T. (2023). Computing a human-like reaction time metric from stable recurrent vision models. Advances in neural information processing systems, 36, 14338-14365.

**Other Comments Or Suggestions:**

NA.

**Other Strengths And Weaknesses:**

NA. Please refer to my above review.

**Questions For Authors:**

I have already conveyed my questions and concerns in other parts of the review. I thank the authors for their submission and look forward to reading their rebuttal.

**Relation To Broader Scientific Literature:**

The work is very relevant to current art in language modeling (LM) with recurrent architectures. Developing LMs with stable hidden state spaces is crucial to efficiently scaling performance at long sequence lengths during inference time. Hence, I find this work to be well situated to broader scientific literature on sequence modeling.

**Theoretical Claims:**

NA.

---

> ### Author Rebuttal · Authors · 2025-04-01
>
> We thank the reviewer for their detailed feedback and helpful discussion around length generalization in recurrent models more broadly. We are encouraged to see that the reviewer believes this type of work is relevant and that our training intervention to the initial states of SSMs are sound. We provide answers to their questions:
>
> > RE: Experiments limited to SSMs and Gated Linear Attention
>
> We understand the reviewer's concern about our results being specific to SSMs and Gated Linear Attention (GLA). **For that reason, we have performed a new experiment in another recurrent architecture, RWKV-v6 [1], and have found a similar phenomenon**: the model trained on short contexts and zero-initialized states diverges, but post-training with state passing enables length generalization:
>
> https://postimg.cc/zLkhvF2x
>
> We will include this result in the final version of the paper to increase the scope of studied recurrent models. We also tried using DeltaNet and Gated DeltaNet [2], but the open source implementations seem to have a bug when using a non-zero initial state (the output of the forward applied to a long context versus substituting part of the context with an initial state do not match), so we could not test our interventions. We hypothesize that the benefits of training with non-zero initialized states will also apply to other modern recurrent networks, but we agree with the reviewer's remark and in the final version of the paper **we will make our claims more specific to the scope of architectures investigated.**
>
> [1] Peng, B. et al. (2024). _Eagle and Finch: RWKV with matrix-valued states and dynamic recurrence_. In _First Conference on Language Modeling_.
>
> [2] Yang, S. et al. _Gated Delta Networks: Improving Mamba2 with Delta Rule_. In _The Thirteenth International Conference on Learning Representations.
>
> > RE: Evaluation on other long context tasks besides passkey retrieval and perplexity
>
> We recognize that passkey retrieval on its own might not be enough to assess the length generalization capabilities of our interventions. Therefore, **we have evaluated the models on two additional long context tasks related to language modeling, which go beyond passkey retrieval in their complexity**. The initial state interventions bring significant benefits in these tasks. Due to limited space, we refer to the response to Reviewer b5dM for more details on the tasks.
>
> Results for the BABILong task: https://postimg.cc/6TjDhBZ0
>
> Results for the synthetic copying task: https://postimg.cc/ctwryLQV
>
> > RE: References related to length extrapolation in recurrent networks and TBTT / state passing
>
> We thank the reviewer for providing these algorithmic extrapolation references. The references propose architecture modifications to enable length extrapolation in recurrent models, and also propose training them on a more diverse distribution of states (Incremental Progress Training). While related, our work is focuses on language modeling, where established models exhibit a diverging perplexity after the context length - which is surprising as this does not require length extrapolation, given that predicting a language token at say position 2500 should be roughly as hard as predicting it at position 2000. Several recent works in language modeling attempt to solve this length generalization issue by changing the inner mechanism of the models (see Section 6 for specific references), so **one of our key contributions resides in showing that length generalization in established language models is achievable through simple (yet underutilized) interventions on the initial state.**
>
> We also agree that some of these interventions were used in prior works, but we believe our main contribution lies in establishing a framework for understanding why they help in recurrent models and providing empirical experiments to support it. In particular, our work: (1) introduces the **"unexplored states hypothesis", which explains the poor generalization performance of recurrent models by reasoning about the distribution of the states attained when processing tokens beyond the training context**; **(2) systematically evaluates a range of interventions on the initial state, including random and fitted noise, which are naturally motivated by the "unexplored states hypothesis"**; and (3) **provides a deeper understanding of the state and how recurrent models processlong context through metrics like Effective Remembrance and the analysis of the results of the interventions**.
>
> Additionally, **we also show that these interventions enable length extrapolation in some tasks related to core aspects of language modeling, like passkey retrieval, synthetic copying and BABILong.** We agree that the algorithmic length extrapolation tasks of the provided references are important for improving machine learning models at large and some of the methods used are similar, so in the final version we will include them both as related work and as avenues for future work.

---

### Decision · Program_Chairs · 2025-05-01

**Decision:**

Accept (poster)

**Comment:**

The paper investigates length generalization in recurrent networks. The authors showed that performance of RNNs drops significantly beyond training context length, and proposed several interventions leading to significant improvement in length generalization. Reviewers appreciate the simple intervention for practical reasons and the empirical validation. Some raised their scores after the rebuttal as their main concerns are addressed. There are some remaining questions, e.g., using perplexity as the measurement, missing discussion/comparison to prior works, but overall reviewers agreed on a more positive outcome, considering the merits outweigh the limitations.